# Do birds of a feather leave the nest together? The role of sibling personality similarity in the transition to adulthood

**Yu-Chin Her** [ORCID]*, **Jorik Vergauwen, Dimitri Mortelmans** [ORCID]

Centre for Population, Family and Health, University of Antwerp, Antwerp, Belgium

* Yu-Chin.Her@uantwerpen.be

**Data Availability Statement:** All datafiles are available from the database of Understanding Society: The UK Household Longitudinal Study

## Abstract

Empirical evidences on intragenerational transmission of life course have been demonstrated and that interpersonal similarity may moderate the effect. In particular, siblings who are more similar in their demographic characteristics are more likely to follow each other's life course transitions. Focusing on parental home-leaving and building upon the social influence processes and similarity-attraction effects, this study investigates whether the association between siblings' departures from the parental home increases when they are similar in the Big Five personality traits, like similarity in demographic traits. We use 28 waves of a longitudinal sample from "Understanding Society: The U.K. Household Longitudinal Study". The results of the multilevel discrete-time event-history analysis ($N$ = 3,717 children) indicate that the association between leaving of a sibling and oneself was strengthened when they had a similar level of extraversion, particularly when they were both introverts. This implies that although introverted adolescents and emerging adults might take less initiative regarding social relationships and be more hesitant in their transition to adulthood, when a similarly introverted sibling makes such a transition, they are more inclined to do so. To conclude, the study uncovers the relationship between siblings' personality similarity and their resemblance in nest-leaving, which helps explain young adults' home-leaving decision in an era when delayed leaving is observed.

## Introduction

The process of nest-leaving has slowed since the 1970s, leading to a delayed transition to adulthood and more years spent at the parental home [1–3], which affects one's further life-course decisions, development, and the parent-child relationship [4, 5]. A number of factors contribute to this delay for young people, including the declining significance of traditional values and the prioritization of lifestyle preferences [2, 6]. At the same time, studies have indicated that one's personality and whether one has a nest-leaving sibling also plays a role in shaping this transition [7, 8].

Existing literature has emphasized the importance of sibling relationships and documented the connectedness between siblings' life decisions and pathways [9, 10]. Siblings are often viewed as lifelong companions and role models [11]. Their intragenerational transmission of

(https://www.understandingsociety.ac.uk), DOI:
10.5255/UKDA–SN-6614-18.

**Funding:** This research has been made possible through the grant Nr. G017519N from the Research Foundation - Flanders (FWO) to Dimitri Mortelmans.

**Competing interests:** The authors have declared that no competing interests exist.

life-course events may not only originate from their shared genes and environment but also from the nature of their interaction with one another [12, 13]. In particular, the process of social influence leads siblings, who are similar in terms of demographic characteristics, to play important roles in each other's decision-making processes [14, 15]. Studies in psychology also suggest that similarity predicts close relationships better than dissimilarity does [16–18].

To date, however, studies focusing on siblings transmission of life courses exclusively looked into demographic similarities and did not examine the potential influence of other traits. In the present study, we draw attention to the role of siblings' Big Five personality traits, particularly the role of sibling similarity in personality with respect to leaving the parental home. Focusing on adolescent siblings and siblings of young adulthood, our goal is to study how and the extent to which siblings' similar personality traits may explain intragenerational transmission of parental-home leaving. By addressing this, our study sheds new light on the existing theories and the relationship between siblings' personality combinations and transition to adulthood.

## Theoretical background

### Leaving the parental home during emerging adulthood

The transition to adulthood, which comprises a series of events that include leaving the parental home, starting a professional career, union formation, and entry into parenthood, has been characterized as "late, protracted, and complex" for the current generation [6]. The timeline of those events' occurrence is lengthening, possibly reversible, and no longer in a certain order. This group of young people (aged 18–29) are referred to as "emerging adults", describing those who feel somewhere in between adolescence and adulthood and who experience instability, role exploration, and change in life pathways [19].

The focus of this study is the transition of leaving the parental nest, which is considered an important step toward establishing a self-sufficient and independent adult life for adolescents and emerging adults. Due to the observed delays in home-leaving and the decreasing normativity of the timing of leaving [1, 20], researchers have focused their attention on investigating the consequences of continued co-residence with parents. First, on-time leavers reported higher rates of romantic activity in comparison to those who did not move out at the same age [21]. Second, those who continued to reside with their parents were less mature and had achieved fewer developmental tasks (e.g., starting a career) [22]. Third, while emerging adults who remain at home may feel as if they are treated like children and show lower levels of well-being [23], their parents are also more likely to feel financially and emotionally burdened [22].

Because of these consequences, we aim to study the reasons that contribute to emerging adults' decision to leave. Studies have indicated that structural constraints (e.g., parental resources and de-standardization of traditional values) [2], demographic features (e.g., gender, education, and employment status) [3, 24, 25], and personality traits [8, 26] have an impact on individuals' timing of leaving the parental home. Social network effects (e.g., siblings, colleagues, friends) may as well be crucial when making life course decisions [12, 27]. For instance, a sibling's leaving was shown to be positively associated with one's own leaving [7]. In this study, we aim to further elaborate on the sibling transmission effect, using siblings' personality similarity.

### Sibling similarity in life-course transitions: Observational learning processes and relationship closeness

Siblings typically live under the same roof and spend a considerable amount of time together during childhood and adolescence [11]. Having their relationship nurtured over years of

exposure, they may have a stronger impact in each other's transition to adulthood compared to other social network members. Previous studies suggest that social influence processes, namely, the processes of social contagion and social learning [12, 28, 29], explain similarities between siblings in terms of their life course trajectories [10]. The theory of *social contagion* suggests that individuals may influence those around them by their behavior, both intentionally and unintentionally. At the same time, people also unconsciously observe and imitate the behaviors of their networks and those they are close to (e.g, age peers, friends, and siblings) [12]. The notion of *social learning* points to the contagiousness of behaviors as well, but has a stronger focus on the observational learning aspect [28]. Similarly, the *paving-the-way hypothesis* posits that siblings who transit to a new chapter pave the way for those who follow [30]. Of relevance to the present study is that the three mechanisms all regard sibling similarity as a driver of mimicking behaviors.

Drawing on these theories, a number of studies have documented that siblings who are alike in terms of demographic characteristics are more likely to be a stronger source of contagion and role models and pave the way for one another. For example, siblings with a smaller age difference have a higher chance of influencing each other regarding dropping out of high school [31], leaving the parental home [7], and entry into parenthood [14]. Same-gender siblings are also more similar with respect to internal migration [30] and patterns of family formation [15]. One of the reasons sibling similarity is related to life-course resemblance is their relationship quality and emotional closeness [13, 31, 32], that siblings similar in age shared more experiences during childhood and could thus enhance relationship closeness in adulthood [33]. Other studies indicated that gender similarity is associated with increased sibling support and interaction [34, 35]. Nevertheless, while we know being similar in demographic traits is an important factor in reinforcing sibling transmission of life course transitions, including nest-leaving, it is unknown whether other forms of sibling similarity function the same.

Next to these social mechanisms, genetics are an important cause of sibling similarity. On average, full siblings inherit 50% of the same genetic variants from their parents, which makes them genetically more predisposed to being similar in their leaving home behavior, among other life course transitions [13, 36, 37]. Likewise, siblings' personality similarity can also be explained by their shared genetics, given that personality traits are to some degree heritable [38, 39]. The focus of our study was to examine which factors moderate intragenerational transmission of leaving, specifically how personality similarity functions as a potential determinant.

## Combing psychological and sociological perspectives: Similarity-attraction effect and personality similarity

Building upon theories of cognitive dissonance and social comparison [40, 41], the similarity-attraction effect (SAE) suggests that individuals favor a logical and consistent view of the world [42]. The perspective suggests that we prefer to encounter people with similarity in personality traits because in doing so, our opinions, ideas, and attitudes are validated. Such reinforcement can be associated with positive feelings and leads to attraction. On the contrary, individuals who have different personality traits often disagree with us and create inconsistency in our worldview. We may thus feel anxious, confused, and repelled by those people. In other words, because similarity affords predictability, it enables individuals to interact in a more relaxed manner. This effect is also at the center of the notion behind homophily, which demonstrates that similarity can serve as a foundation for interpersonal attraction [43] and that relationship between dissimilar individuals dissolve more quickly [44]. Despite mixed

empirical evidence [45], most literature suggests that personality similarity is linked to better interpersonal relationships than is personality dissimilarity [18, 46].

While the previously discussed sociological theories emphasize that similarity in demographics facilitates warm relationships, the SAE stresses the effect of similar personality traits. Research even showed that personality offers a stronger similarity effect than demographic characteristics do in both Canadian and Japanese contexts [18]. Moreover, the SAE may account for a great deal of social contagion and peer influence [47]. Taken together, it seems that the sociological and psychological theories share a commonality, that similar individuals tend to have increased transmission and contagion of behaviors, likely via enhanced interaction and close relationship between them. When it comes to cross-sibling effects on nest-leaving, it is important to examine whether and how siblings' personality similarity, serving as a proxy for their relationship closeness, strengthens the intragenerational effect of leaving the parental home.

## The role of sibling personality similarity in transmission of leaving the parental home

It is widely accepted that a five-factor model of personality (the Big Five) encompasses the most salient aspects of personality [48] that shape individuals' behaviors and experiences [49]. *Extraversion* is characterized by being enthusiastic, assertive, and positive, and the tendency to enjoy and be dominant in social interaction. *Agreeableness* refers to the tendency to engage in prosocial behaviors and to value harmony, friendliness, and positive relationships. *Neuroticism* is defined by being vulnerable to stress and not being able to cope with negative emotions. Finally, *openness to experience* is expressed in the appreciation of art, unconventional ideas/ creativity, curiosity, and intellect. Regarding the similarity effect of the Big Five model, there is consensus that the traits of extraversion, agreeableness and neuroticism are most relevant to people's social behavior [46, 50, 51]. In the following, we discuss how siblings' Big Five traits can be related to similarity in their timing of home-leaving.

As for *extraversion*, scholars found that dyads who have the same level of extraversion (both introverts or both extraverts) interact more easily than dyads composed of an introvert and an extravert do [46]. This suggests that the interaction style between people with the same personality makes their relationship more enjoyable. In a similar vein, people often get along with those showing the same level of extraversion because they share the same amount of sociability [16, 17]. Importantly, it was particularly dyads in which both are introverts that had higher odds of being friends and having joyful interactions [46, 52]. This suggestion goes hand in hand with the aforementioned social influence processes that similar demographic characteristics, associated with enhanced relationship quality, lead to life-course resemblance. Because better interaction and relationship closeness may increase the likelihood that a sibling's leaving paves the road, Hypothesis 1 (H1) was formed: The association between siblings' departures is stronger when they have similar levels of extraversion (a) and, particularly, when they are both introverts (b).

*Agreeableness* is often characterized by the ability to control undesirable emotions, develop healthy and harmonious relationships, and be less involved in aggressive, threatening, or conflict-related behaviors [53, 54]. In terms of homophily by agreeableness, studies have shown that adolescents and emerging adults are inclined to form friendships with and be closest to those who have a similar level of agreeableness [16, 17]. They demonstrated that two agreeable people form the most beneficial relationship, compared to a disagreeable pair or a pair in which one is agreeable and the other is disagreeable. Concerning the role of agreeableness in sibling relationships, agreeableness is one of the best predictors of high-quality sibling

relationships [55]. In order to have a warmer and less conflictual sibling relationship, it is important for siblings to have high levels of agreeableness. Considering this, Hypothesis 2 (H2) is as follows: The association between siblings' departures is stronger when they both have high levels of agreeableness.

While agreeableness is related to healthy and close sibling relationships [55], *neuroticism* is negatively associated with the quality of sibling relationships [55–57]. Concerning the similarity-attraction effect, similarity in neuroticism can encourage friendship and group formation [58]. Individuals who are similarly non-neurotic are more inclined to be friends, because conflicts in dyads may escalate when both members are emotionally unstable [52]. When homophily is manifested in neuroticism, it seems reasonable that when siblings are both not neurotic, their relationship and interactions are more pleasant and harmonious. As a result, we formed Hypothesis 3 (H3): The association between siblings' departures is stronger when they both have low levels of neuroticism.

*Openness to experience* has traditionally been considered an intrapsychic characteristic which pertains to individuals' intellectual lives and is less associated with their social behavior and relationships, unlike extraversion, agreeableness, and neuroticism [50, 59]. However, similarity in openness can also predict whether individuals become friends [16]. That is, although openness is not a particularly meaningful trait for interpersonal connection, adolescents' self-rated openness tends to be similar to those of their ideal friends [60]. The aforementioned studies did not indicate whether the effect of personality similarity is especially strong at only one end of the trait dimension, whereas the other demonstrated that only those with greater openness have similar friends [52]. Based on these studies, Hypothesis 4 (H4) suggests the following: The association between siblings' departures is stronger when they have similar levels of openness (a), and, particularly, when they both have high levels of openness (b).

## Data and methods

### Dataset selection and structure

The data for this longitudinal study are drawn from the survey Understanding Society: The U. K. Household Longitudinal Study (UKHLS) [61]. UKHLS is an ongoing annual and nationally representative panel survey that gathers a variety of information at the individual and family level. The dataset consists of 28 waves; waves 1 to 18 belong to the British Household Panel Survey (BHPS) and waves 19 to 28 are from Understanding Society. Together, the waves cover the entire period between 1991 and 2019. Understanding Society is a continued version of the BHPS, featuring data harmonized between both studies, allowing us to track households across the whole period. Over 60% of those who participated in BHPS decided to continue with Understanding Society [61].

To capture sibling effects of personality traits in function of parental home leaving, we considered all children from a family, unlike previous studies selecting families with only two children when studying sibling transmission [10, 62]. This approach results in a more representative sample of multiple-child families. To model the associations between all considered siblings, we created the following structure. As shown in Fig 1, our data structure consists of four levels: family level (level 4), child level (level 3), sibling dyad level (level 2), and wave/time level (level 1). The time level is nested in the sibling dyad level, which is nested in the child level, embedded in the family level. Because we considered all children from a family, the data structure is more complex compared to a design with only two children per family (as those are mirroring siblings). Therefore, the sibling dyad level is added to the model. Fig 1 provides an example with three children (X, Y and Z) at risk of leaving the household. Each of them can function both as a child at risk and as a sibling with potential influence on another

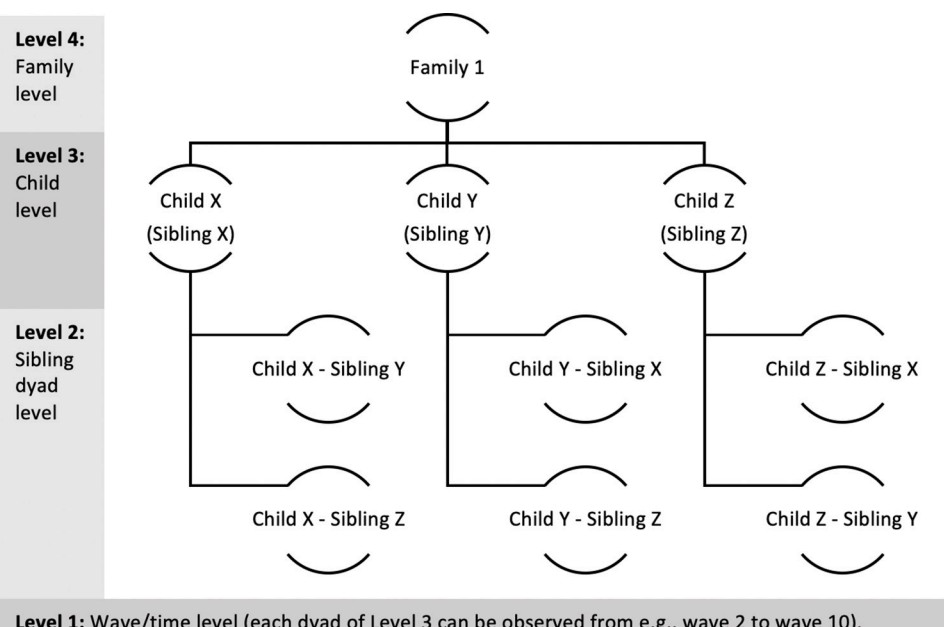

**Fig 1. Example of the four-level data structure.**

sibling's departure. For instance, when Child X is at risk, Child Y and Child Z are the siblings of influence (i.e., Sibling Y and Sibling Z), leading to sibling dyads XY and XZ. If Child Y is at risk, Child X and Child Z become the mirroring siblings (i.e., Sibling X and Sibling Z), resulting in sibling dyads YX and YZ. At the wave/time level the absence or occurrence of an event is measured for the child at risk.

Siblings were identified based on the mother's identification, in case of biological siblings. To take into account children from blended families, we included all children joining the household from either the mother's or the father's side in case of a parental dissolution. To determine whether children had left the parental home, we relied on the information of the household grid. In other words, when they were no longer registered in the household grid, they were considered to have left the nest. There were several conditions in which disappearing from the household was not counted as leaving, ensuring that children who were no longer reported in the household grid most likely left the parental household. First, when children left the household because of parental relationship dissolution, as long as they still lived with one of their biological parents, they were not regarded as having left but as new members of blended families. Second, children who passed away were right censored. Third, if a family stopped participating in the survey or migrated outside of the United Kingdom, attrition of the whole family was right censored as well.

Children in some families had already left the nest prior to participating in the survey. To avoid bias, we did not take those families into account. Finally, we left censored those children who had left the parental home before the age of 16. Given our focus on the departure of adolescents and emerging adults from the parental home, those who had not left the nest at the age of 29 were right censored. When children switched roles to the sibling of potential influence, they were required to be at least 13 years old. There was no maximum age restriction for the siblings of potential influence. Overall, we analyzed 33,612 observations in 4,976 sibling dyads (representing 3,717 children and 1,845 families).

## Measures

**Dependent variable.** Our dependent variable was an indicator reflecting whether or not the child at risk has left the parental home, translating into a conditional probability documenting the risk of the children living in the household to leave at any given time [63]. In total, 53.38% of the children at risk had an event, and the mean age at the time of leaving was 22.31 (*SD* = 3.15). Censored children left the observation window at age 24.68 on average (*SD* = 3.48). While they might have experienced an event further, this was not captured in our selected data. At the age of 29, 335 children did not leave the parental home.

**Variables of interest.** At-risk children's siblings may or may not be living in the household when they had an event. To indicate whether having a nest-leaving sibling is associated with one's own leaving, we created a variable *sibling left*, which is a dichotomous and time-varying variable at the sibling dyad level. When the dyadic sibling of the at-risk child had an event, we assigned the at-risk child a value of 1 (0 = sibling had not left). To examine how the association between siblings' departures may be strengthened or weakened depending on the child's and the sibling's personality traits, we used this indicator to interact with the Big Five personality scales.

The Big Five personality traits were measured in waves 15 and 21 of the UKHLS, and among children aged 16 or above [61]. When children functioned as siblings of potential influence, their personality traits were taken as siblings' personality traits. The mean age of children was 19.77 (*SD* = 3.19) in wave 15 and 20.56 (*SD* = 3.33) in wave 21. Whenever information on the Big Five was available, we took the information and imputed it for all the waves, making the Big Five variables time-constant. When they were measured in both waves, we used the information of the last wave, given that personality becomes more stable throughout the life span [64, 65]. A majority of the literature showed that personality does not reach stability before age 25 and is still changing during adulthood [65]. For this reason, for children who provided the personality information in both waves, we tested the extent to which their personality changed. We found that the change was minimal, with around 75% of the observations showing at most a one-point change. This reassured us that treating the personality variables time-constant did not substantially bias the results when looking at personality during young adulthood and its association with nest leaving.

Each of the Big Five traits was measured using three items on a 7-point scale (1 = does not apply to me at all, 7 = applies to me perfectly) [61]. The mean score of the items was taken as the final score for a particular trait. Some examples of the items are "has a forgiving nature" for extraversion, "does things efficiently" for conscientiousness, "is talkative" for agreeableness, "values artistic and aesthetic experiences" for openness, and "worries a lot" for neuroticism. The higher someone scored on a particular trait, the better the trait described that person, and vice versa.

To test the moderating role of siblings' personality similarity, we calculated the absolute difference between the personality traits of the siblings [18, 66, 67] and subsequently reversed the values, resulting in five variables: similarity in extraversion, similarity in conscientiousness, similarity in agreeableness, similarity in openness, and similarity in neuroticism. Those variables were time-constant with values ranging between 0 and 6. The higher the score was on this indicator, the more similar their traits were. The larger the difference, the more similar the child was to their sibling.

**Control variables.** We controlled for a number of background characteristics that were known to impact the timing of children's nest-leaving [3, 25, 68]. Those covariates were included at the family level, the child level, or the dyad level. First, at the family level, parents' educational background reflected the highest educational qualification achieved. It was

grouped into four categories: high (have a degree), middle (completed A-level or secondary high school), low (everything below the other two categories), and unknown. Employment status of the parents was categorized as either being an employee/self-employed, no employment, or unknown employment. The parents' marital/relationship status could either be in a relationship (including being married or cohabiting with a partner) or single (i.e., divorced or separated). An unknown category was included to avoid losing a considerable number of observations due to missing values. For the parental covariates, when both biological parents were present, we selected the mother. Otherwise, we took the available biological parent as the targeted parent. The number of children each family had was included as a continuous variable.

Second, at the child level, age was modelled using both a linear and a quadratic term. Gender was a dummy variable (ref = male). Children's educational attainment, employment status, and relationship status were measured using the same categories as for the parents. Third, because we also considered siblings of blended families, we controlled for whether the sibling was a child's biological sibling at the dyad level by including a dummy indicator (ref = not a biological sibling). Moreover, we controlled for the age difference between siblings in its absolute value (e.g., an age difference of 2 or -2 were both counted as 2) to account for departing around the same time due to age similarity. Table 1 provides an overview of the descriptive statistics at each level. Whether these covariates were time variant or invariant is also indicated in Table 1.

## Analytical strategy

The analysis drew on discrete-time event-history models predicting the timing of the at-risk children's nest-leaving using a logit link function. The models were estimated by the PROC GLIMMIX procedure in SAS 9.4. We applied multiple strategies to adjust for potential sources of confounding. Besides a causal effect of someone's leaving on their sibling via social learning or contagion, nest-leaving could also represent the effects of different endogenous factors. For instance, because siblings are raised in the same family context, they experience similar family norms and expectations of when to leave the parental home [69, 70]. As a result, the models included random effects (RE) at the family, child, and sibling dyad levels. Moreover, to control for household background, the models introduced various socioeconomic and demographic characteristics as covariates, aiming to avoid the impact of confounders on our results [9, 15]. The analysis indicated that allowing random intercepts at the different levels resulted in an absence of unexplained variance at the sibling dyad level. Therefore, as a first robustness check, we repeated the analysis by randomly selecting one sibling dyad per child (i.e., 3,717 sibling dyads for 3,717 children).

A second robustness check repeated the analysis by using fixed-effects (FE) models. Since at-risk children and their siblings stem from the same families, it is plausible that children from the same family simply leave the nest around the same age or time because they share a number of unobserved/endogenous characteristics (e.g., parental expectations, family norms) [69]. FE models reduce heterogeneity biases originating from time-invariant family influences [71] and are complementary to RE models, which are more generalizable [72]. Therefore, we draw on the FE models to address confounding by household background factors. The robustness checks can be found in the supplementary analyses (see the S1 File).

## Results

### Main analysis

Table 2 presents the results of the event-history analysis (the full table is in the Table S1 in S1 File). All Big Five personality traits were modeled simultaneously when examining the main

**Table 1. Descriptive statistics of the dependent and independent variables.**

| Variables | N | % | M | SD |
|---|---|---|---|---|
| **Outcome variable at the child level (*N* = 3,717)** | | | | |
| Event * | | | | |
| Yes | 1,984 | 53.38% | | |
| No | 1,733 | 46.42% | | |
| **Family/parental level characteristics (*N* = 1,845)** | | | | |
| Educational level | | | | |
| High | 426 | 23.09% | | |
| Middle | 754 | 40.87% | | |
| Low (ref.) | 582 | 31.54% | | |
| Unknown | 83 | 4.50% | | |
| Employment status * | | | | |
| Full-time/part-time | 1,101 | 59.67% | | |
| No employment (ref.) | 552 | 29.92% | | |
| Unknown | 192 | 10.41% | | |
| Relationship status * | | | | |
| Married/cohabiting/has a partner | 771 | 41.79% | | |
| Divorced/separated/single (ref.) | 869 | 47.10% | | |
| Unknown | 205 | 11.11% | | |
| Number of children | | | 2.90 | 1.16 |
| **Child level characteristics (*N* = 3,717)** | | | | |
| Age * | | | 23.52 | 3.53 |
| Gender | | | | |
| Male (ref.) | 1,841 | 49.53% | | |
| Female | 1,876 | 50.47% | | |
| Educational level * | | | | |
| High | 1,000 | 26.90% | | |
| Middle | 1,846 | 49.66% | | |
| Low (ref.) | 227 | 6.11% | | |
| Unknown | 644 | 17.33% | | |
| Employment * | | | | |
| Full-time/part-time | 2,075 | 55.82% | | |
| No employment (ref.) | 1,003 | 26.98% | | |
| Unknown | 639 | 17.19% | | |
| Relationship status * | | | | |
| Married/cohabiting/has a partner | 92 | 2.48% | | |
| Divorced/separated/single (ref.) | 2,762 | 76.66% | | |
| Unknown | 832 | 22.38% | | |
| Child's Big Five traits (range: 1–7) | | | | |
| Extraversion | | | 4.74 | 1.14 |
| Conscientiousness | | | 5.02 | 1.07 |
| Agreeableness | | | 5.45 | 1.02 |
| Openness | | | 4.74 | 1.21 |
| Neuroticism | | | 3.76 | 1.35 |
| **Sibling dyad level characteristics (*N* = 4,976)** | | | | |
| Biological sibling | | | | |
| Yes | 4,759 | 95.64% | | |
| No (ref.) | 217 | 4.36% | | |

(*Continued*)

**Table 1.** (Continued)

| Variables | N | % | M | SD |
|---|---|---|---|---|
| Siblings' age difference | | | 3.60 | 2.70 |
| Sibling left * | | | | |
| Yes | 2,224 | 44.69% | | |
| No (ref.) | 2,752 | 55.31% | | |
| Sibling's Big Five traits (range: 1–7) | | | | |
| Extraversion | | | 4.71 | 1.14 |
| Conscientiousness | | | 5.09 | 1.06 |
| Agreeableness | | | 5.46 | 1.02 |
| Openness | | | 4.73 | 1.22 |
| Neuroticism | | | 3.76 | 1.36 |
| Sibling similarity in the Big Five (range: 0–6) | | | | |
| Similarity in extraversion | | | 4.81 | 0.93 |
| Similarity in conscientiousness | | | 4.90 | 0.87 |
| Similarity in agreeableness | | | 4.95 | 0.87 |
| Similarity in openness | | | 4.76 | 1.00 |
| Similarity in neuroticism | | | 4.56 | 1.11 |

*Note*: Descriptive statistics were presented at each level. For the time-variant variables (*), we present the descriptive statistics of the last observed wave.

and interaction effects. To avoid a long table, interaction terms that were not significant or relevant to the hypotheses were not reported in Table 2 but can be found in Table S1 in S1 File.

The models were estimated stepwise. In Model 1, we included background characteristics at the family, child, and sibling dyad levels, together with the Big Five traits of the at-risk children and the siblings. The model showed that at-risk children whose parents had a higher educational background and were divorced/separated were more likely to leave the parental home. The number of children in the family was not related to the leaving of the parental home. At the child level, female, employed, and non-single children showed a higher tendency to leave. Children with a middle level of education had a higher chance than those with a lower educational background of remaining at home. At the sibling dyad level, a positive association between siblings' departures was observed, indicating that an association between siblings' timing of leaving. The at-risk children were less likely to leave when they had a biological bond with their sibling and when there was a larger age difference between them. In terms of the Big Five personality traits of the children at risk, our results suggest that among all the Big Five traits, only openness to experience had a statistically significant effect in that the more open one was, the less likely one was to leave. We did not find any significant associations between a sibling's Big Five traits and the timing of a child's leaving.

Model 2 tested whether siblings' similarity in the Big Five traits was related to their similarity in the timing of nest-leaving by including the sibling personality similarity variables and the terms of interaction with sibling left. To better understand the interaction effects, we discussed the results by means of predicted probability plots [73]. As shown in Model 2 and Fig 2, the more similar the siblings were regarding their level of extraversion, the more likely a child was to leave following their sibling's departure ($b = 0.144$, $SE = 0.064$). In other words, sibling similarity in extraversion was positively associated with similarity in their timing of leaving the parental home, confirming H1a. When a sibling did not leave the nest, similarity in extraversion was linked to a longer stay in the parental home. Although the effect of openness was not statistically significant ($b = -0.103$, $SE = 0.060$), we observed from the plotted interaction of

**Table 2. Unstandardized coefficients of the multilevel discrete-time event-history analysis predicting at-risk children's event of leaving (*N* time level = 33,612).**

| | Model 1 | | Model 2 | | Model 3 | |
|---|---|---|---|---|---|---|
| | b | se | b | se | b | se |
| Intercept | -21.980 | 1.317*** | -21.532 | 1.338*** | -24.059 | 2.047*** |
| Age | 1.423 | 0.109*** | 1.422 | 0.109*** | 1.416 | 0.109*** |
| Age$^2$ | -0.023 | 0.002*** | -0.023 | 0.002*** | -0.023 | 0.002*** |
| *Family/parental level characteristics* | | | | | | |
| Education (ref: low) | | | | | | |
| High | 0.624 | 0.132*** | 0.618 | 0.131*** | 0.621 | 0.132*** |
| Middle | 0.301 | 0.112** | 0.292 | 0.111** | 0.297 | 0.111** |
| Unknown | -0.240 | 0.237 | -0.219 | 0.236 | -0.206 | 0.236 |
| Employment (ref: no) | | | | | | |
| Yes | 0.045 | 0.084 | 0.055 | 0.084 | 0.051 | 0.084 |
| Unknown | -0.912 | 0.228*** | -0.906 | 0.228*** | -0.889 | 0.228*** |
| Relationship status (ref: divorced/separated) | | | | | | |
| Married/cohabiting | -0.181 | 0.081* | -0.179 | 0.081* | -0.180 | 0.080* |
| Unknown | -0.576 | 0.210** | -0.580 | 0.210** | -0.594 | 0.210** |
| Number of children | 0.010 | 0.038 | 0.016 | 0.038 | 0.011 | 0.038 |
| *Child level characteristics* | | | | | | |
| Gender (ref: male) | 0.648 | 0.091*** | 0.645 | 0.090*** | 0.643 | 0.091*** |
| Education (ref: low) | | | | | | |
| High | -0.204 | 0.159 | -0.207 | 0.159 | -0.205 | 0.159 |
| Middle | -0.296 | 0.142* | -0.301 | 0.141* | -0.300 | 0.141* |
| Unknown | -0.394 | 0.278 | -0.399 | 0.277 | -0.394 | 0.277 |
| Employment (ref: no) | | | | | | |
| Yes | 0.152 | 0.069* | 0.153 | 0.069* | 0.149 | 0.069* |
| Unknown | -0.420 | 0.247 | -0.419 | 0.247 | -0.428 | 0.246 |
| Relationship status (ref: single) | | | | | | |
| In a relationship | 0.718 | 0.225** | 0.721 | 0.225** | 0.734 | 0.226** |
| Unknown | 0.627 | 0.118*** | 0.631 | 0.118*** | 0.628 | 0.118*** |
| Child's Big Five traits | | | | | | |
| Extraversion | 0.060 | 0.040 | 0.061 | 0.040 | 0.445 | 0.171** |
| Conscientiousness | 0.034 | 0.043 | 0.046 | 0.043 | 0.155 | 0.199 |
| Agreeableness | -0.055 | 0.045 | -0.064 | 0.046 | -0.116 | 0.217 |
| Openness | -0.116 | 0.038** | -0.120 | 0.038** | -0.268 | 0.130* |
| Neuroticism | 0.054 | 0.034 | 0.051 | 0.034 | 0.208 | 0.094* |
| *Sibling dyad level characteristics* | | | | | | |
| Biological sibling (ref: no) | -0.563 | 0.186** | -0.542 | 0.185** | -0.573 | 0.185** |
| Siblings' age difference | -0.042 | 0.013** | -0.042 | 0.013** | -0.043 | 0.013** |
| Sibling left (ref: no) | 0.741 | 0.063*** | 0.025 | 0.569 | 3.597 | 2.256 |
| Sibling's Big Five traits | | | | | | |
| Extraversion | -0.007 | 0.031 | | | 0.404 | 0.169* |
| Conscientiousness | -0.034 | 0.035 | | | 0.039 | 0.198 |
| Agreeableness | 0.010 | 0.036 | | | -0.041 | 0.216 |
| Openness | -0.046 | 0.029 | | | -0.197 | 0.127 |
| Neuroticism | -0.014 | 0.025 | | | 0.111 | 0.096 |
| Sibling similarity in extraversion | | | -0.094 | 0.047* | | |
| Sibling similarity in openness | | | 0.036 | 0.045 | | |
| *2-way Interactions* | | | | | | |

(*Continued*)

**Table 2.** (Continued)

| | Model 1 | | Model 2 | | Model 3 | |
|---|---|---|---|---|---|---|
| | **b** | **se** | **b** | **se** | **b** | **se** |
| Sibling similarity in E x Sibling left | | | 0.144 | 0.064* | | |
| Sibling similarity in O x Sibling left | | | -0.103 | 0.060 | | |
| Child's E x Sibling's E | | | | | -0.073 | 0.035* |
| Child's E x Sibling left | | | | | -0.582 | 0.227* |
| Sibling's E x Sibling left | | | | | -0.620 | 0.227** |
| *3-way interactions* | | | | | | |
| Child's E x Sibling's E x Sibling left | | | | | 0.102 | 0.046* |
| Unexplained variances family level | 0.798 | 0.149*** | 0.777 | 0.148*** | 0.759 | 0.146*** |
| Unexplained variances child level | 2.980 | 0.222*** | 2.953 | 0.221*** | 2.957 | 0.222*** |

*Note*: Big Five traits were all modeled together. The effect of sibling similarity in the other Big Five traits as well as the 2-way interactions and 3-way interactions of the other traits can be found in Table S1 in the S1 File.

* p< .05

** p< .01

*** p< .001.

predicted probabilities (Fig 3) that, if siblings were dissimilar in their level of openness, the effect of sibling left was stronger, which contradicts H4a.

In Model 3, we further examined whether the effect of personality similarity among siblings may differ depending on the level of extraversion, agreeableness, openness, and neuroticism. It was possible that the association between siblings' departures is only strengthened or weakened on one end of the spectrum (e.g., both of them are highly agreeable). To test this, three-way interaction effects were modeled using at-risk children's Big Five traits, those of their siblings, and the indicator of whether the sibling had left or not. As shown in Model 3 and Fig 4, a significant relationship between sibling left, the at-risk child's level of extraversion, and the sibling's level of extraversion was found ($b = 0.102$, $SE = 0.046$). When the sibling of an at-risk

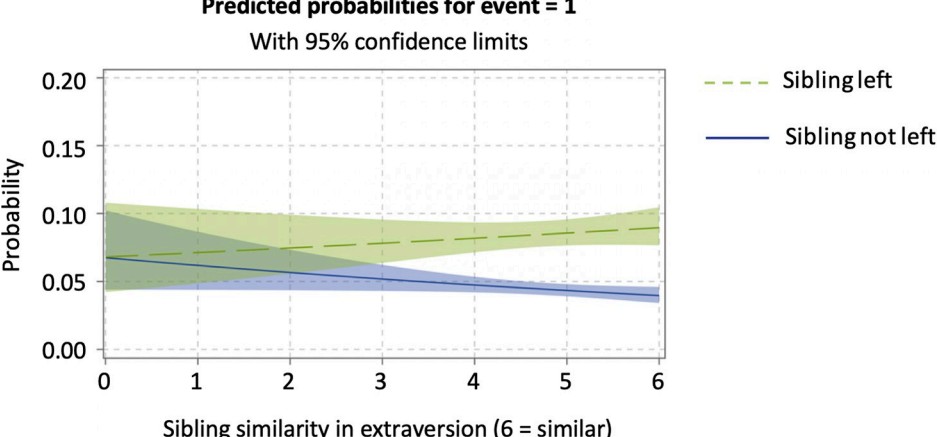

**Fig 2. The association between siblings' departures as moderated by siblings' similarity in extraversion.** The *x*-axis represents how siblings are similar in terms of their level of extraversion, and a value of 6 indicates that they are highly similar.

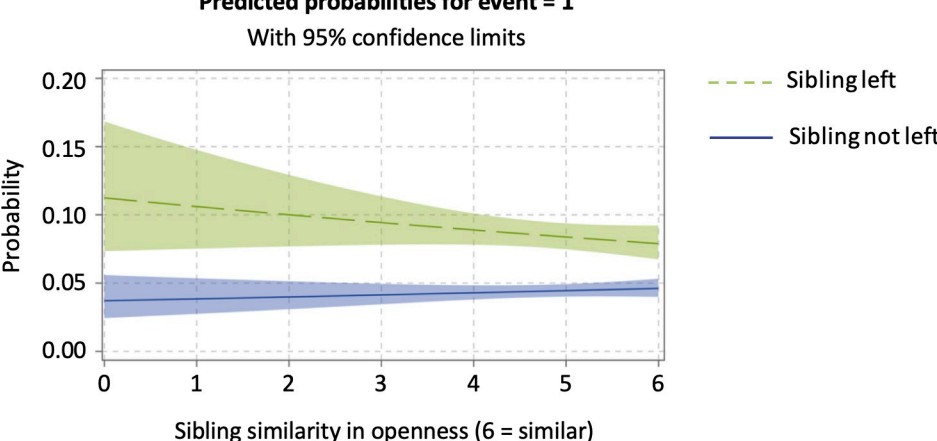

**Fig 3. The association between siblings' departures as moderated by siblings' similarity in openness.** The *x*-axis represents how siblings are similar in terms of their level of openness, and a value of 6 indicates that they are highly similar.

child had left the parental home, the child was most likely to leave when he/she was an introvert and when the sibling was also an introvert. To understand what introversion entails in this case, we explored different introversion cut-off scores. We found that introverted siblings were more likely to leave home together when they both were above average introverts. If the sibling was introverted and the child was extraverted, the likelihood of leaving decreased.

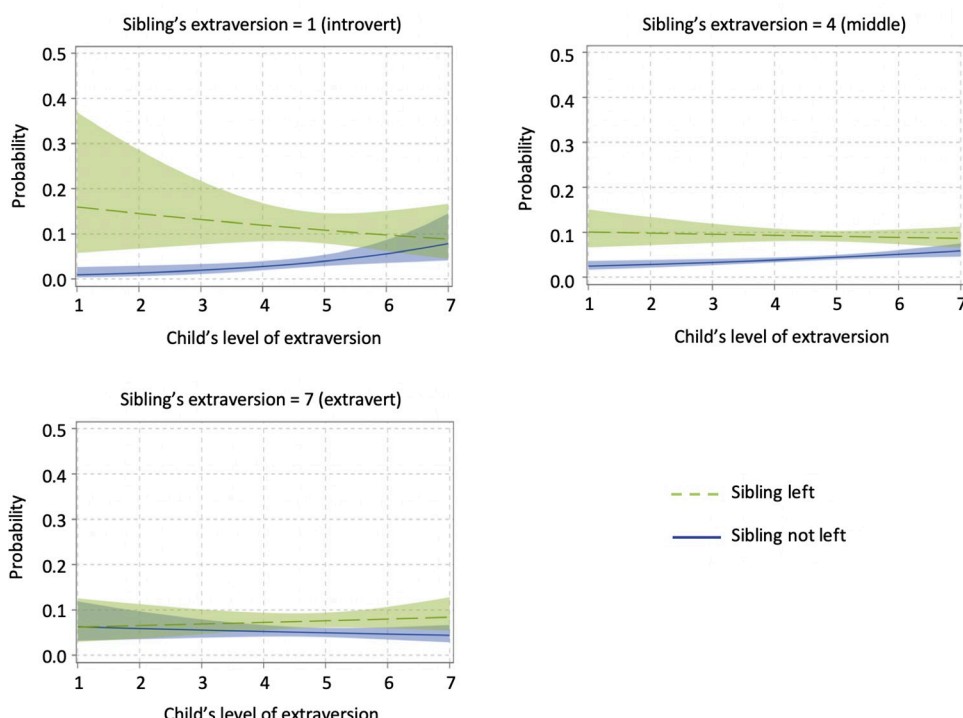

**Fig 4. The association between siblings' departures as moderated by child's extraversion and sibling's extraversion.** (Predicted probabilities for event = 1, with 95% confidence limits).

Similarly, introverted children were not motivated by their extraverted siblings' departure. When both of them were extraverted, the probability of leaving was higher than in dissimilar pairs, but still lower than for two introverts. This supports H1b that the relationship between siblings' departures is particularly strong when they are both introverts. In addition, if both the child at risk and the sibling lived at home, the child's risk of departure was the highest if they were extraverted and the sibling was introverted. Given that we did not find a significant three-way interaction effect for agreeableness, neuroticism, and openness, H2, H3, and H4b were not confirmed.

### Robustness check and supplementary analysis

For the first robustness check, the models estimating a three-level data structure (excluding sibling dyad level) yielded almost identical results compared to the main analysis (see Table S2 in the S1 File). For the second robustness check using FE models, although the intercept of sibling left was shifted to a lower level and its effect became negative, the extent to which similarity in siblings' personality moderated the effect was similar to that in Model 2. The three-way interaction effects in the FE model were also in line with those in Model 3 (see Table S3 in the S1 File).

As discussed, agreeableness predicts better interpersonal relationships. To further ensure that the effects found in the main analysis were not confounded by sibling relationship quality, we repeated the analysis based on sub-samples of agreeable and unagreeable dyads (agreeableness scores = 5 being the cutoff point; see supplementary analysis 4 in the S1 File). For both agreeable and unagreeable sibling dyads, the interaction effect of sibling similarity in extraversion was close to the significant level of 0.05, which is likely due to the smaller sample size. However, the direction and magnitude of the effect remained the same, that sibling similarity in extraversion positively moderated sibling transmission of parental home leaving. As for the three-way interaction effect of extraversion, the results of agreeable dyads showed high resemblance to those of Model 3 in Table 2, whereas no effect was found for unagreeable dyads.

Furthermore, we controlled for parental agreeableness, a proxy for parent-child relationship quality, and regions in which the family resided on a reduced sample as a sensitivity check (see supplementary analysis 5 in the S1 File). They largely mimicked the main analysis, confirming that our findings are robust. Finally, we explored whether there was any non-linear effects of the Big Five traits. The results can be found in Fig S1 in the S1 File.

### Conclusion and discussion

Previous research has focused on studying intragenerational transmission of life course trajectories using siblings' demographic characteristics, such as gender composition, birth order, and age difference [15, 62]. However, no study has yet explored whether similarity in siblings' personalities may function in the same way as similar demographic characteristics, having an impact on the timing of their home-leaving. Based on the observational learning process and the similarity-attraction theory, it is likely that siblings who left the parental home first pave the way for those who follow and especially when they have a similar personality.

Our findings suggest that a sibling's departure was more transmissible when they shared a similar level of extraversion, particularly when they were both introverted. This echoes previous studies that a similar rather than dissimilar degree of extraversion is connected to better social interactions and friendship formation [16, 46]. It is also in line with the theory of social contagion that individuals who are more similar tend to have a higher impact on each other's life course decisions [7, 12, 14, 15]. Siblings who are both introverts may be even more likely to form an influential source on leaving because of the more predictable, understandable, and

enjoyable introverted attraction [46, 52]. This implies that, although introverted adolescents and young adults might take less initiative in social relationships and be more hesitant to make important decisions, when a similarly introverted sibling makes such a transition, they are more inclined to do so as well. We additionally observed that, when a sibling did not leave, dissimilarity in extraversion facilitated the at-risk child's leaving. In this case, children at risk might leave early because they are annoyed by their dissimilar sibling residing at home. However, our third sensitivity analysis tentatively suggests that the three-way interaction results might be mediated by agreeableness and/or sibling relationship quality, as highly agreeable sibling-dyads seemed to be driving the finding. This needs further examination with more optimal measure of sibling relationship quality. Altogether, extraversion seems to be related to siblings' interaction and how contagious their adulthood transition is to one another, given that siblings' agreeableness were accounted for in all models and that all other robustness checks confirmed this finding.

As for sibling similarity in openness, although the effect was not significant, having a nest-leaving sibling appeared to delay one's own leaving when a similar level of openness was observed, unlike what was found with regard to extraversion. This may hint that individuals are more attracted to and impacted by their siblings who are the opposite of themselves or complementary in terms of openness. Although agreeableness and neuroticism are often correlated with interpersonal relationships, we did not find empirical evidence that the association between siblings' departures can be explained by sibling similarity in these two traits. This could mean that, despite the fact that being neurotic and not agreeable were negatively associated with homophily (e.g., a higher risk of relationships dissolution, less chance of a harmonious sibling relationship) [56, 74], they do not seem to change the strength of sibling contagion with regard to nest-leaving.

Being similar in personality and timing of leaving to one's siblings can be due to both genetic and environmental influences. To which degree associations between the home leaving of siblings can be interpreted as sibling influence rather than the consequence of another shared environmental or genetic factor is difficult. The longitudinal design of our study explicitly models the order of events (at-risk child leaving after sibling had already left) and takes into account both random and fixed family effects (in separated models), thus correcting for shared genetics, among others. This allows a more confident interpretation of the estimates as sibling influence. However, as we cannot rule out that the results are driven by genetic effects completely, the alternative interpretation should be considered as well. If both the sibling effect and personality similarity are completely due to genetic similarity, the identified effects in this paper are suggestive of genetic correlations between the phenotypes home leaving and personality [75–77]. In other words, the results would suggest that pleiotropic genetic effects are present, which affect both home leaving and personality and cause similarity among siblings for both phenotypes.

To our knowledge, this study is the first to examine the process of nest-leaving using both one's own personality traits and those of one's sibling. It is also the first to study the association between siblings' personality similarity and the resemblance of their life course trajectories. In both the main analysis and the robustness check, we were able to take into account all siblings from a family in a multilevel structure and follow them for up to 14 years. Therefore, our findings are applicable to families with more than two children. In conclusion, our findings suggest that whether the similarity-attraction hypothesis provides support and explanation for sibling resemblance in leaving may depend on the personality trait in question. Among all the hypothesized traits, sibling similarity in extraversion was best linked to attraction and mimicking behaviors, like siblings belonging to the same gender and having a similar age. Parents and social programs supporting transitions to adulthood for adolescents and emerging adults

could benefit from the study and understand why some children leave and others stay when their sibling has or has not left. Children's early/delayed leaving can also be assisted better by knowing how siblings' personality combinations are linked to their mimicking behaviors with regard to life course transitions.

Some limitations must be taken into account when interpreting the study results. First, with the current data, we could not directly assess whether siblings' personality similarity facilitated the association between their nest-leaving because of their relationship closeness. While, in line with the literature, we anticipated that sibling similarity leads to enhanced sibling relationships, which, in turn, yields stronger contagion effects, this mechanism needs to be tested in detail. Future studies should investigate whether relationship closeness truly mediates the observed effects. Second, even though we found little change in the personality traits for those who filled in the information in more than one wave, the use of time-constant personality was suboptimal. Especially for those who left the parental home in their late 20s and reported on the personality measured around age 20, their personality might be different from what was captured earlier. Replicating the study with time-varying personality information is thus highly encouraged. Third, because of the observational nature of the study, even though the family-level FE models were included as a robustness check, we still cannot completely exclude confounding by unobserved variables, limiting causal interpretations of the results. Besides addressing these limitations, it might be important for future research to study the association between siblings' personality similarity and their similarity in the timing of transitioning to other life courses (e.g., union formation and dissolution). Furthermore, qualitative analysis exploring the reasons adolescents and young adults are more influenced by siblings with the same level of extraversion may provide a deeper understanding of the topic.

## Supporting information

**S1 File.**
(DOCX)

## Author Contributions

**Conceptualization:** Yu-Chin Her, Jorik Vergauwen, Dimitri Mortelmans.

**Data curation:** Yu-Chin Her, Dimitri Mortelmans.

**Formal analysis:** Yu-Chin Her.

**Funding acquisition:** Jorik Vergauwen, Dimitri Mortelmans.

**Methodology:** Yu-Chin Her, Jorik Vergauwen, Dimitri Mortelmans.

**Project administration:** Jorik Vergauwen, Dimitri Mortelmans.

**Supervision:** Jorik Vergauwen, Dimitri Mortelmans.

**Visualization:** Yu-Chin Her.

**Writing – original draft:** Yu-Chin Her, Jorik Vergauwen.

**Writing – review & editing:** Yu-Chin Her, Jorik Vergauwen.

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
