## [Decision Letter · Decision Letter 0]

2 Feb 2023

PONE-D-22-27514Do birds of a feather leave the nest together? The role of sibling personality similarity in transition to adulthoodPLOS ONE

Dear Dr. Her,

Thank you for submitting your manuscript to PLOS ONE. After careful consideration, we feel that it has merit but does not fully meet PLOS ONE’s publication criteria as it currently stands. Therefore, we invite you to submit a revised version of the manuscript.

No questions can be added to an already collected data set. So try to get as much from available data as possible, while making sure the analysis won't be challenged like the article on ESG investment in the most recent New Scholar podcast.

https://www.youtube.com/watch?v=LbIBa7R5Yeg

In other words, please try to make sure that there is no need to publish an article on the same topic from the same data set again.

Reviewers raised a relevant question regarding the relationship (personal (1) and genetic (2)) between siblings. Please give it a thought.

With regards to Reviewer 1, my first idea would be to check whether the results stay the same when considering only siblings in a good relationship - a good relationship could be approximated as two highly agreeable siblings (let's say to 25%). An extended idea is to look up also high-low, low-high (leaving in the othe opposite order), and low-low settings. But the decision is yours. Of course, you will be able to compare only 4 BFI personality traits then.

With regards to Reviewer 2, it sounds reasonable to try also non-linear functional forms. Maybe you will be able to brainstorm additional ideas after reading the comments.

You will likely need to extend the theory section because of additional modeling.

Irrespective of reviewers' comments, please conclude only facts, and do not try to overgeneralize.

We look forward to receiving your revised manuscript.

Kind regards,

Frantisek Sudzina

Academic Editor

PLOS ONE

Journal Requirements:

2. We note that you have referenced (ie. Bewick et al. [5]) which has currently not yet been accepted for publication. Please remove this from your References and amend this to state in the body of your manuscript: (ie “Bewick et al. [Unpublished]”) as detailed online in our guide for authors

Reviewers' comments:

Reviewer's Responses to Questions

**Comments to the Author**

1. Is the manuscript technically sound, and do the data support the conclusions?

Reviewer #1: Yes

Reviewer #2: Yes

2. Has the statistical analysis been performed appropriately and rigorously? 

Reviewer #1: I Don't Know

Reviewer #2: I Don't Know

3. Have the authors made all data underlying the findings in their manuscript fully available?

Reviewer #1: Yes

Reviewer #2: Yes

4. Is the manuscript presented in an intelligible fashion and written in standard English?

Reviewer #1: Yes

Reviewer #2: Yes

5. Review Comments to the Author

Reviewer #1: This article studies how sibling similarity in their personality is associated to sibling nest leaving. The authors theorize on sibling influence, the behavior of one sibling affecting the other, and they use ‘paving the way’ arguments, hypothesizing that siblings who are more similar in their personality are more likely to move out once the first sibling has moved out. The authors conclude that similarity in extraversion, especially when both are introverts, was associated to stronger associations between one’s own nest leaving and that of the sibling.

Main points

The authors clearly explain the importance of timely nest leaving for adolescents, but why is it important to study dyadic nest leaving? Why would siblings with similar personalities leave together? I understand that influence plays a role, but given the dyadic approach in this paper, I would have expected some more reasoning as to why sibling nest leaving is not a matter of age or simply the sum of the decision of two individuals, or a result of the personality of two individuals. Are these decisions truly that dependent? I think the paper could be improved in the introduction/background and discussion section by stressing the importance and contribution of studying dyadic nest leaving more.

Before one could argue that similar siblings “leave the nest together” it seems important to control for the quality of the sibling relationship as well as the quality of the intergenerational relationship, i.e., siblings liking each other may postpone moving out (together) or when they do not like each other enjoy time home alone with their parents. Siblings not being close to their parents may be encouraged to leave the nest early regardless of their (similar) personalities. Lastly, could you control for the distance to the first sibling moving out? Effects may differ when parents live in a rural area and the first sibling moves out to a large university city or when parents were already living in such a city and the first sibling moves out moves to a student flat nearby.

It was not clear to me whether this data set is a multi-actor dataset. Are the personality traits etc. reported by both siblings? If the data are multi-actor, are the reports of both siblings first considered as ‘child’ and next as ‘sibling’? Related to this is the question how should I understand the child at risk? Does this mean that only children are considered of which at least one sibling has already left the household? In that case the reports of both siblings cannot be used once as child and once as sibling. While some effort has certainly already been made, I would like some more explanation of the data structure and implications for analysis.

The authors argue that similarity on extraversion especially holds for both siblings being introverts (page 19-20). From which numbers do the authors derive this conclusion? Moreover, from what score on extraversion are you considered to be an introvert? Perhaps robustness checks could be performed dichotomizing this variable on the introvert cut-off score to see whether the results hold.

Smaller points

• The abstract starts with a reference to social network studies, but this is not quite the focus of the article. Therefore, the second sentence does not logically follow from this first sentence.

• In the background section some references are made to homophily and social network studies, but some major references to social network studies about homophily are lacking, such as

McPherson, M., Smith-Lovin, L., & Cook, J. M. (2001). Birds of a feather: Homophily in social networks. Annual Review of Sociology, 27, 415–444. https://doi.org/10.1146/annurev.soc.27.1.415

• “Siblings were identified based on mother’s lineage” (page 10). What does that mean?

• The structure of the sample description (page 9-11) could be improved. My suggestion would be to first discuss the inclusion criteria, then all decisions made and to end with the N of the analytical sample.

• “… those who had not left the nest at the age of 29 were right censored” (page 11). How many observations did you lose with this decision?

• Related to this at page 11: “53.8% of the children at risk had an event”. Does this mean that 46% still lives at home? What age are they?

• Page 13: “… we selected the mother. Otherwise, we took the available biological parent as the targeted parent”. How often does this occur? Are there any implications of this decision?

• Why are the main effects of sibling personality not reported in model 2?

• Instead of reporting the full table with all two-way and three-way interactions in the main text, the authors could consider to report a slimmed version of the table and to report the extended version in an online Appendix. I am no expert on these analyses, but would it for example be an option to report linear combinations of coefficients instead of the interactions?

• Sometimes you refer to odds (e.g., page 20 “the odds of leaving decreased”), whereas b’s are reported in table 2. Is this correct? Could you show the SE’s in addition to the significance stars?

• Figure 1: this figure is not mentioned in the text. Is the description of level 1 correct? It refers to the dyad level at level 3, but shouldn’t this be level 2?

Reviewer #2: The study addresses an interesting question: Is the association between siblings’ departures from the parental home increased when siblings are similar in some Big Five personality traits, particularly with respect to specific poles of the trait dimension? The authors reported evidence for extraversion.

All in all, the paper is well written and sounds convincing. The authors provided lots of environmental sibling interaction theories that are useful to account for the siblings’ similarity in traits and departures from parental home. However, this results in several blind spots that reduced my enthusiasm for the current paper.

First, the authors appear to be completely blind for the fact that siblings are genetically related and that personality traits are at least to some degree heritable (e.g., Kandler & Papendick, 2017; Vukasovic & Bratko, 2015). As a consequence, siblings share on average 50% of their genes that can vary among humans. Thus, they can be more or less genetically similar than 50%. More genetically similar siblings may decide more similarly, such as leaving parental home at the same time of life. All the hypotheses can be completely and thus alternatively framed by gene-environment correlations and transactions (see Briley et al., 2018, 2019; Kandler et al., 2019).

Second, it is definitely not the case that age 16 is the age at which personality starts to be stable. There are different stability indices and none of them show stability after age 16 (see Bleidorn et al., 2022).

Furthermore, even though I am not familiar with the analytical strategy that was used, it seems to be straightforward to me. The figures are quite helpful. The analyses, however, seem to rely on the linearity assumption within and across levels as well as for main and interaction effects. Given the relatively good sample size, would it not be possible to model nonlinear effects, additionally? This may help to unravel further interesting trends that may or may not in line with your hypotheses. I was thinking about response surface analyses with the siblings’ trait scores as the two predictors for the similarity of departure. This would be a completely other approach, but would allow linear and nonlinear as well as interaction effects. Anyway, I think that it might be worth considering nonlinearity to get more out of the data.

In sum, the study is interesting and should be published. However, the authors should revise their paper and do a better job in really combing and combining psychological and sociological perspectives by taking genetic factors into account. They should also do a better job in embedding their study into the broad literature on personality development.

Thanks for asking me to read this interesting study.

I wish the authors good luck with their paper,

Christian Kandler

References

Bleidorn, W., Schwaba, T., Zheng, A., Hopwood, C. J., Sosa, S. S., Roberts, B. W., & Briley, D. A. (2022). Personality stability and change: A meta-analysis of longitudinal studies. Psychological Bulletin, 148(7-8), 588–619. https://doi.org/10.1037/bul0000365

Briley, D. A., Livengood, J., & Derringer, J. (2018). Behavior genetic frameworks of causal reasoning for personality psychology. European Journal of Personality, 32, 202–220. http://dx.doi.org/10.1002/per.2153

Briley, D. A., Livengood, J., Derringer, J., Tucker-Drob, E. M., Fraley, R. C., & Roberts, B. W. (2019). Interpreting behavior genetic models: Seven developmental processes to understand. Behavior Genetics, 49, 196–210. http://dx.doi.org/10.1007/s10519-018-9939-6

Kandler, C., & Papendick, M. (2017). Behavior genetics and personality development: A methodological and meta-analytic review. In J. Specht (Ed.), Personality development across the lifespan (pp. 473–495). San Diego, CA: Elsevier Academic Press. http://dx.doi.org/10.1016/B978-0-12-804674-6.00029-6

Kandler, C., Waaktaar, T., Mõttus, R., Riemann, R., & Torgersen, S. (2019). Unravelling the interplay between genetic and environmental contributions in the unfolding of personality differences from early adolescence to young adulthood. European Journal of Personality, 33,

221–244. http://dx.doi.org/10.1002/per.2189

Vukasovic, T., & Bratko, D. (2015). Heritability of personality: A meta-analysis of behavior genetic studies. Psychological Bulletin, 141, 769–785. http://dx.doi.org/10.1037/bul0000017

6. PLOS authors have the option to publish the peer review history of their article (what does this mean?). If published, this will include your full peer review and any attached files.

Reviewer #1: No

Reviewer #2: **Yes: **Christian Kandler

---

## [Author Response · Author response to Decision Letter 0]

16 Mar 2023

Dear editor,

Thank you for the opportunity to submit a revision of our manuscript, “Do Birds of a Feather Leave the Nest Together? The Role of Sibling Personality Similarity in Transition to Adulthood”. We appreciate the time taken by the editor and reviewers to provide constructive feedback on the manuscript and have done our best to address their concerns.

We carefully reviewed the comments and revised the manuscript accordingly. We have done some robustness checks to address e.g., 1) the comments on whether our findings were confounded by inter-/intra-generational relationship quality using the agreeableness measures, and 2) the comments suggesting to try out the non-linear terms, among others. 

Below, we have provided numbered point-by-point responses to the comments from the reviewers and indicated the places where changes were made. We also made some small language edits and fixed some typos. Track changes were used in the revised manuscript.

Comments to the authors:

Reviewer 1:

This article studies how sibling similarity in their personality is associated to sibling nest leaving. The authors theorize on sibling influence, the behavior of one sibling affecting the other, and they use ‘paving the way’ arguments, hypothesizing that siblings who are more similar in their personality are more likely to move out once the first sibling has moved out. The authors conclude that similarity in extraversion, especially when both are introverts, was associated to stronger associations between one’s own nest leaving and that of the sibling.

Main points:

1. Comments: 

The authors clearly explain the importance of timely nest leaving for adolescents, but why is it important to study dyadic nest leaving? Why would siblings with similar personalities leave together? I understand that influence plays a role, but given the dyadic approach in this paper, I would have expected some more reasoning as to why sibling nest leaving is not a matter of age or simply the sum of the decision of two individuals, or a result of the personality of two individuals. Are these decisions truly that dependent? I think the paper could be improved in the introduction/background and discussion section by stressing the importance and contribution of studying dyadic nest leaving more.

Response:

Thank you for the suggestion. We agree that parental home leaving is indeed related to children’s age. This was accounted for by the covariates: at-risk child’s age, the quadratic term of at-risk child’s age, and siblings’ age gap. It may also be associated with one’s own personality traits, as accounted for in the regression models and discussed in the introduction on page 3:

“At the same time, studies have indicated that one’s personality and whether one has a nest-leaving sibling also play a role in shaping this transition (7, 8).”

And in the literature review on pages 4-5:

“Studies have indicated that structural constraints (e.g., parental resources and de-standardization of traditional values, 2), demographic features (e.g., gender, education, and employment status, 3, 24, 25), and personality traits (e.g., 8, 26) have an impact on individuals’ timing of leaving the parental home.”

While the focus of this paper is on the effect of siblings’ personality similarity and we do not deny impact from other factors, we agree that we did not explain our decision well as for why we focused on sibling transmission on nest leaving. We now elaborated more in the literature review why studying sibling influence is of importance. 

On page 5:

“Social network effects (e.g., siblings, colleagues, friends) may as well be crucial when making life course decisions (12, 27). For instance, a sibling’s leaving was shown to be positively associated with one’s own leaving (7). In this study, we aim to further elaborate on the sibling transmission effect, using siblings’ personality similarity.”

“Siblings typically live under the same roof and spend a considerable amount of time together during childhood and adolescence (11). Having their relationship nurtured over years of exposure, they may have a stronger impact in each other’s transition to adulthood compared to other social networks.”

The intention of the paper is to look at mechanisms behind the social interaction and social network effects (e.g., sibling as an important social network) on leaving the parental home. The dyadic structure is one of most optimal ways that we think can help us to investigate this. The method can be applied to children with more than one sibling (which is what we did and one of our main contributions), allowing us to look at how each sibling’s personality plays a role, as stated on page 10:

“To capture sibling effects of personality traits in function of parental home leaving, we considered all children from a family, unlike previous studies selecting families with only two children when studying sibling transmission (10, 62). This approach results in a more representative sample of multiple-child families. To model the associations between all considered siblings, we created the following structure. As shown in Fig 1, our data structure consists of four levels: family level (level 4), child level (level 3), sibling dyad level (level 2), and wave/time level (level 1)..[…]”

2. Comments: 

Before one could argue that similar siblings “leave the nest together” it seems important to control for the quality of the sibling relationship as well as the quality of the intergenerational relationship, i.e., siblings liking each other may postpone moving out (together) or when they do not like each other enjoy time home alone with their parents. Siblings not being close to their parents may be encouraged to leave the nest early regardless of their (similar) personalities. Lastly, could you control for the distance to the first sibling moving out? Effects may differ when parents live in a rural area and the first sibling moves out to a large university city or when parents were already living in such a city and the first sibling moves out moves to a student flat nearby.

Response:

Thank you very much for the comment. We agree that sibling relationship quality and parent-child relationship quality are both important for one’s parental home leaving decision. However, with the current dataset, it is difficult to take them into account. Sibling relationship quality and parent-child relationship quality are measured in the youth data of UKHLS. Yet, the youth questionnaires were only sent to children aged between 10 and 15, making it impossible to include them in our sample. (It is reasonable to look at parental home leaving for those aged 16 and above, and questions about the big five personality traits were also only asked among those who aged 16 and above, as mentioned in the paper.) 

In the original version of the manuscript, we have mentioned not being able to examine directly the effect of sibling relationship as a limitation on page 24:

“First, with the current data, we could not directly assess whether siblings’ personality similarity facilitated the association between their nest-leaving because of their relationship closeness. While, in line with the literature, we anticipated that sibling similarity leads to enhanced sibling relationships, which, in turn, yields stronger contagion effects, this mechanism needs to be tested in detail. Future studies should investigate whether relationship closeness truly mediates the observed effects.”

Questions related to parent-child relationship quality were also asked in the main questionnaires. Despite this, they were only asked in a few waves and a large amount of missing values in the variables were observed. Combining the available information with the current sample, we ended up with very few complete cases to conduct our analyses. 

However, as discussed in the literature review, studies have consistently shown that being agreeable is associated with a good interpersonal relationship. 

In the current analysis, we controlled for agreeableness of both the at-risk children and their siblings and did not find a contribution of the variables: In Model 1, both variables on agreeableness were not significant. In Model 3, the interaction terms were not significant. This means that being agreeable oneself and having an agreeable sibling, signaling a good sibling relationship was not associated with similarity in leaving the parental home. 

To further examine the robustness of the significant effect of extraversion and to make sure that it was not confounded by agreeableness or sibling relationship quality, we repeated the analysis based on sub-samples of agreeable and unagreeable dyads (agreeableness scores = 5 being the cutoff point; see the additional analysis 3 in Appendices). For both agreeable and unagreeable sibling dyads, the interaction effect of sibling similarity in extraversion was close to the significant level of 0.05, which is likely due to the smaller sample size. However, the direction and magnitude of the effect remained the same, that sibling similarity in extraversion positively moderated sibling transmission of parental home leaving. As for the three-way interaction effect of extraversion, the results of agreeable dyads showed high resemblance to those of Model 3 in Table 2, whereas no effect was found for unagreeable dyads. This hints that the three-way interaction results might be mediated by agreeableness and/or sibling relationship quality, as highly agreeable sibling-dyads seemed to be driving the finding. However, this needs further examination with more optimal measure of sibling relationship quality. Altogether, we believe that extraversion is nevertheless related to siblings’ interaction and how contagious their adulthood transition is to one another, given that siblings’ agreeableness were accounted for in all models and that all other robustness checks confirmed this finding. We have illustrated the results in both the manuscript and Appendices and discussed the possible implications in the discussion. 

In the main text, we added on page 21:

“As discussed, agreeableness predicts better interpersonal relationships. To further ensure that the effects found in the main analysis were not confounded by sibling relationship quality, we repeated the analysis based on sub-samples of agreeable and unagreeable dyads (agreeableness scores = 5 being the cutoff point; see the additional analysis 3 in Appendices). For both agreeable and unagreeable sibling dyads, the interaction effect of sibling similarity in extraversion was close to the significant level of 0.05, which is likely due to the smaller sample size. However, the direction and magnitude of the effect remained the same, that sibling similarity in extraversion positively moderated sibling transmission of parental home leaving. As for the three-way interaction effect of extraversion, the results of agreeable dyads showed high resemblance to those of Model 3 in Table 2, whereas no effect was found for unagreeable dyads.”

In the discussion on pages 22-23:

“However, our third sensitivity analysis tentatively suggests that the three-way interaction results might be mediated by agreeableness and/or sibling relationship quality, as highly agreeable sibling-dyads seemed to be driving the finding. This needs further examination with more optimal measure of sibling relationship quality. Altogether, extraversion seems to be related to siblings’ interaction and how contagious their adulthood transition is to one another, given that siblings’ agreeableness were accounted for in all models and that all other robustness checks confirmed this finding.”

In Appendices on page 5, we added Table A4 and the following:

“To further test whether the interaction effects of extraversion were confounded by agreeableness, a proxy of relationship quality, we stratified our sample based on siblings’ agreeableness and repeated Model 2 and 3 in Table 2. When both the at-risk children and their siblings scored 5 or above on agreeableness, they were counted as agreeable sibling dyads. When they scored below 5, they represented the unagreeable dyads.”

For the same reason as for sibling relationship quality, we additionally controlled for parental level of agreeableness in an additional analysis. This did not change the main findings. The results can be found in Table A5 in the Appendices. 

 We do not have the information on the distance to the first sibling who moved out in the data. The information of living in a rural or urban area is only available in Understanding Society and not BHPS (i.e., from wave 19 onwards in our data). However, we agree with the reviewer that where the children reside with their parents can be important. As a result, we controlled for the region (the UK government office region) of the parental home. Adding this covariate did not change the results (see Table A5 in the Appendices). 

The results with parental agreeableness and region was not added to the main analysis but added as additional robustness checks for two reasons. First, our sample size would be reduced substantially. Second, given the current number of modeled covariates and interaction terms, we believe it is better to not over complicate the models.

 For the two additional covariates, we wrote on page 21 of the main text:

“Furthermore, we controlled for parental agreeableness, a proxy for parent-child relationship quality, and regions in which the family resided on a reduced sample as a sensitivity check (see additional analysis 4 in Appendices). They largely mimicked the main analysis, confirming that our findings are robust.”

 And on page 6 of Appendices:

“After accounting for parental agreeableness (proxy for parent-child relationship quality) and official regions in the UK in which the family resided, the effects found in Table 2 were still present (see Table A5 below).”

3. Comments: 

It was not clear to me whether this data set is a multi-actor dataset. Are the personality traits etc. reported by both siblings? If the data are multi-actor, are the reports of both siblings first considered as ‘child’ and next as ‘sibling’? Related to this is the question how should I understand the child at risk? Does this mean that only children are considered of which at least one sibling has already left the household? In that case the reports of both siblings cannot be used once as child and once as sibling. While some effort has certainly already been made, I would like some more explanation of the data structure and implications for analysis.

Response:

Indeed, personality traits were reported by all children. All children can function both as children at risk and siblings whenever they met the selection criteria. We now made it clearer on page 12:

“The Big Five personality traits were measured in waves 15 and 21 of the UKHLS, and among children aged 16 or above (61). When children functioned as siblings of potential influence, their personality traits were taken as siblings’ personality traits.”

Children were at risk as long as they reached the age 16 and still reside in the household. They can be at risk even though their siblings had not left the household. This was made clearer on page 12: 

“At-risk children’s siblings may or may not be living in the household when they had an event. To indicate whether having a nest-leaving sibling is associated with one’s own leaving, we created a variable sibling left, which is a dichotomous and time-varying variable at the sibling dyad level. When the dyadic sibling of the at-risk child had an event, we assigned the at-risk child a value of 1 (0 = sibling had not left).”

4. Comments: 

The authors argue that similarity on extraversion especially holds for both siblings being introverts (page 19-20). From which numbers do the authors derive this conclusion? Moreover, from what score on extraversion are you considered to be an introvert? Perhaps robustness checks could be performed dichotomizing this variable on the introvert cut-off score to see whether the results hold.

Response:

Thank you for the comment. It is definitely a good idea to check for cut-off points to see in what situation the interaction effect occurred. We explored different introversion cut-off scores and found that introverted siblings were more likely to leave home together when they both were above average introverts.

On page 20 we indicated that: 

“To understand what introversion entails in this case, we explored different introversion cut-off scores. We found that introverted siblings were more likely to leave home together when they both were above average introverts.”

Smaller points:

5. Comments: 

The abstract starts with a reference to social network studies, but this is not quite the focus of the article. Therefore, the second sentence does not logically follow from this first sentence.

Response:

Thank you for pointing this out. We have changed the first sentence of the abstract on page 2 to the following:

“Empirical evidences on intragenerational transmission of life course have been demonstrated and that interpersonal similarity may moderate the effect.”

6. Comments: 

In the background section some references are made to homophily and social network studies, but some major references to social network studies about homophily are lacking, such as

McPherson, M., Smith-Lovin, L., & Cook, J. M. (2001). Birds of a feather: Homophily in social networks. Annual Review of Sociology, 27, 415–444. https://doi.org/10.1146/annurev.soc.27.1.415

Response: 

Thank you for the suggestion. We indeed missed this important literature. It has been added now to the manuscript on page 7:

“This effect is also at the center of the notion behind homophily, which demonstrates that similarity can serve as a foundation for interpersonal attraction (43) and that relationship between dissimilar individuals dissolve more quickly (44).”

7. Comments: 

“Siblings were identified based on mother’s lineage” (page 10). What does that mean?

Response: 

It means mother’s ID. We agree that the term is a bit unclear and have now changed it to mother’s identification on page 11:

“Siblings were identified based on the mother’s identification, in case of biological siblings.”

8. Comments: 

The structure of the sample description (page 9-11) could be improved. My suggestion would be to first discuss the inclusion criteria, then all decisions made and to end with the N of the analytical sample.

Response: 

Thank you for the suggestion. We have restructured the data structure section. We now shifted the following part to the second paragraph of the data and method section on pages 10-11:

“To capture sibling effects of personality traits in function of parental home leaving, we considered all children from a family, unlike previous studies selecting families with only two children when studying sibling transmission (10, 62). This approach results in a more representative sample of multiple-child families. To model the associations between all considered siblings, we created the following structure. As shown in Fig 1, our data structure consists of four levels: family level (level 4), child level (level 3), sibling dyad level (level 2), and wave/time level (level 1). The time level is nested in the sibling dyad level, which is nested in the child level, embedded in the family level. Because we considered all children from a family, the data structure is more complex compared to a design with only two children per family (as those are mirroring siblings). Therefore, the sibling dyad level is added to the model. Fig 1 provides an example with three children (X, Y and Z) at risk of leaving the household. Each of them can function both as a child at risk and as a sibling with potential influence on another sibling’s departure. For instance, when Child X is at risk, Child Y and Child Z are the siblings of influence (i.e., Sibling Y and Sibling Z), leading to sibling dyads XY and XZ. If Child Y is at risk, Child X and Child Z become the mirroring siblings (i.e., Sibling X and Sibling Z), resulting in sibling dyads YX and YZ. At the wave/time level the absence or occurrence of an event is measured for the child at risk.”

9. Comments: 

 “… those who had not left the nest at the age of 29 were right censored” (page 11). How many observations did you lose with this decision?

Response: 

Thank you for the question. At age 29, 335 observed children did not have an event. We added this information to the manuscript on page 12:

“At the age of 29, 335 children did not leave the parental home.”

10. Comments: 

Related to this at page 11: “53.8% of the children at risk had an event”. Does this mean that 46% still lives at home? What age are they?

Response: 

Yes, this is correct. We added this to the text and corrected a typo on page 12:

“In total, 53.38% of the children at risk had an event, and the mean age at the time of leaving was 22.31 (SD = 3.15). Censored children left the observation window at age 24.68 on average (SD = 3.48). While they might have experienced an event further, this was not captured in our selected data.”

11. Comments: 

Page 13: “… we selected the mother. Otherwise, we took the available biological parent as the targeted parent”. How often does this occur? Are there any implications of this decision?

Response: 

In 3/4 of the cases, we had information on both biological parents and selected the mother’s dataline. In the other cases, we took the available parent’s information, or randomly selected one parent in case of same-gender couples (around 0.3%). While we are aware that paternal demographic factors were not accounted for in the analysis, we believe that this decision would not have had an impact on the results. Our main interest is to look at the factors that may drive siblings to follow each other’s path of nest leaving. The children’s education and employment were controlled for in the models, which are more important on their decision to leave the parental home. In additional, the parental characteristics we controlled for still help us understand greatly intergenerational transmission from parental SES. Moreover, the effect of parental education was taken into account by the family level fixed effects in our second robustness test.

12. Comments: 

Why are the main effects of sibling personality not reported in model 2?

Response: 

The main effects of sibling personality were not reported in Model 2 because we believe that by including personality of the at-risk children and siblings’ personality similarity, sibling personality was accounted for altogether.

13. Comments: 

Instead of reporting the full table with all two-way and three-way interactions in the main text, the authors could consider to report a slimmed version of the table and to report the extended version in an online Appendix. I am no expert on these analyses, but would it for example be an option to report linear combinations of coefficients instead of the interactions?

Response: 

We thank the reviewer for the suggestion and we agree that Table 2 was a bit lengthy. We now trimmed down Table 2 by reporting only the interaction terms of extraversion and openness in the table and added the full table to Table A1 in the Appendices. We did not exclude the results of the control variables as we believe there are some relevant significant results for getting a better picture of what impact parental home leaving. In all the other tables in the Appendices (Table A2-A5) we reported only the main effect and interaction terms related to extraversion, as they are the main findings we aim to do robustness checks for. 

On page 17, we added:

“Table 2 presents the results of the event-history analysis (the full table is in Appendices as Table A1). All Big Five personality traits were modeled simultaneously when examining the main and interaction effects. To avoid a long table, interaction terms that were not significant or relevant to the hypotheses were not reported in Table 2 but can be found in Table A1.”

14. Comments: 

Sometimes you refer to odds (e.g., page 20 “the odds of leaving decreased”), whereas b’s are reported in table 2. Is this correct? Could you show the SE’s in addition to the significance stars?

Response: 

It is correct that we did not report odds ratio but unstandardized beta, in which positive coefficients mean increased probability of leaving and negative estimates mean decreased probability of leaving. To avoid misunderstanding, we now removed the wording “odds” when discussing the results. We also agree with the reviewer to report SEs additionally. In all regression tables, both in the main text and in the Appendices, SEs are included.

On page 20:

“If the sibling was introverted and the child was extraverted, the likelihood of leaving decreased.”

“In addition, if both the child at risk and the sibling lived at home, the child’s risk of departure was the highest if they were extraverted and the sibling was introverted.”

15. Comments: 

Figure 1: this figure is not mentioned in the text. Is the description of level 1 correct? It refers to the dyad level at level 3, but shouldn’t this be level 2?

Response: 

Figure 1 was mentioned in the main text as Fig 1. The sibling dyad level is indeed level 2 (as mentioned in Fig 1). We made it more explicit now on pages 10-11:

“To capture sibling effects of personality traits in function of parental home leaving, we considered all children from a family, unlike previous studies selecting families with only two children when studying sibling transmission (10, 62). This approach results in a more representative sample of multiple-child families. To model the associations between all considered siblings, we created the following structure. As shown in Fig 1, our data structure consists of four levels: family level (level 4), child level (level 3), sibling dyad level (level 2), and wave/time level (level 1). The time level is nested in the sibling dyad level, which is nested in the child level, embedded in the family level. Because we considered all children from a family, the data structure is more complex compared to a design with only two children per family (as those are mirroring siblings). Therefore, the sibling dyad level is added to the model. Fig 1 provides an example with three children (X, Y and Z) at risk of leaving the household. Each of them can function both as a child at risk and as a sibling with potential influence on another sibling’s departure. For instance, when Child X is at risk, Child Y and Child Z are the siblings of influence (i.e., Sibling Y and Sibling Z), leading to sibling dyads XY and XZ. If Child Y is at risk, Child X and Child Z become the mirroring siblings (i.e., Sibling X and Sibling Z), resulting in sibling dyads YX and YZ. At the wave/time level the absence or occurrence of an event is measured for the child at risk.”

Reviewer 2:

The study addresses an interesting question: Is the association between siblings’ departures from the parental home increased when siblings are similar in some Big Five personality traits, particularly with respect to specific poles of the trait dimension? The authors reported evidence for extraversion. All in all, the paper is well written and sounds convincing. The authors provided lots of environmental sibling interaction theories that are useful to account for the siblings’ similarity in traits and departures from parental home. However, this results in several blind spots that reduced my enthusiasm for the current paper.

1. Comments: 

First, the authors appear to be completely blind for the fact that siblings are genetically related and that personality traits are at least to some degree heritable (e.g., Kandler & Papendick, 2017; Vukasovic & Bratko, 2015). As a consequence, siblings share on average 50% of their genes that can vary among humans. Thus, they can be more or less genetically similar than 50%. More genetically similar siblings may decide more similarly, such as leaving parental home at the same time of life. All the hypotheses can be completely and thus alternatively framed by gene-environment correlations and transactions (see Briley et al., 2018, 2019; Kandler et al., 2019).

Response:

We thank the reviewer for this important point and agree that genetic interpretations need to be discussed. It was not our intention, to create the impression that similarities between siblings must be solely due to environmental factors. 

The longitudinal design of our study helps estimating sibling influences on parental home leaving, which are more likely due to environmental influences. By explicitly modeling the order of events (at-risk child leaving after sibling already had left) and taking into account the random and fixed effects of family (which included common genetic effects), the sibling influence estimates are less likely to represent shared genetics. In other words, we are not merely modeling the co-occurrence of home leaving at similar ages. 

The focus of our study was to examine which factors moderate this sibling influence and we identified patterns of personality as a potential determinant. As correctly pointed out, these personality combinations are the result of both genetic and environmental influences. Even if the personality combinations were completely genetically driven, this would not change the conclusion, that the personality phenotypes interact with sibling influences. While we believe, that the sibling effects represent mostly environmental effects, it may be also worth discussing the scenario, that the sibling influence and similarity in personality are completely due to similar genetics among siblings. In this scenario, the identified effects in this paper are suggestive of a genetic correlations between the phenotypes home leaving and personality. 

In the original manuscript, we briefly mentioned the role of shared genes on page 3: 

“Their intragenerational transmission of life-course events may originate from not only their shared genes and environment but also from the nature of their interaction with one another (12, 13).”

We agree that it should be discussed more in detail. Using the suggested literature, we acknowledged the genetic influence further on page. 6:

“Next to these social mechanisms, genetics are an important cause of sibling similarity. On average, full siblings inherit 50% of the same genetic variants from their parents, which makes them genetically more predisposed to being similar in their leaving home behavior, among other life course transitions (13, 36, 37). Likewise, siblings’ personality similarity can also be explained by their shared genetics, given that personality traits are to some degree heritable (38, 39). The focus of our study was to examine which factors moderate intragenerational transmission of leaving, specifically how personality similarity functions as a potential determinant.”

We also elaborated on our assumptions and different interpretations in the discussion on pages 23-24:

“Being similar in personality and timing of leaving to one's siblings can be due to both genetic and environmental influences. To which degree associations between the home leaving of siblings can be interpreted as sibling influence rather than the consequence of another shared environmental or genetic factor is difficult. The longitudinal design of our study explicitly models the order of events (at-risk child leaving after sibling had already left) and takes into account both random and fixed family effects (in separated models), thus correcting for shared genetics, among others. This allows a more confident interpretation of the estimates as sibling influence. However, as we cannot rule out that the results are driven by genetic effects completely, the alternative interpretation should be considered as well. If both the sibling effect and personality similarity are completely due to genetic similarity, the identified effects in this paper are suggestive of genetic correlations between the phenotypes home leaving and personality (76-78). In other words, the results would suggest that pleiotropic genetic effects are present, which affect both home leaving and personality and cause similarity among siblings for both phenotypes.” 

2. Comments: 

Second, it is definitely not the case that age 16 is the age at which personality starts to be stable. There are different stability indices and none of them show stability after age 16 (see Bleidorn et al., 2022).

Response:

Thank you for your comment. We agree that personality often does not start to be stable around the age of 16. We referred to the study by Bleidorn et al. (2022) in the revised manuscript and revised the paragraph on the personality in the method section on pages 12-13:

“The Big Five personality traits were measured in waves 15 and 21 of the UKHLS, and among children aged 16 or above (61). When children functioned as siblings of potential influence, their personality traits were taken as siblings’ personality traits. The mean age of children in was 19.77 (SD = 3.19) in wave 15 and 20.56 (SD = 3.33) in wave 21. Whenever information on the Big Five was available, we took the information and imputed it for all the waves, making the Big Five variables time-constant. When they were measured in both waves, we used the information of the last wave, given that personality becomes more stable throughout the life span (64, 65). A majority of the literature showed that personality does not reach stability before age 25 and is still changing during adulthood (65). For this reason, for children who provided the personality information in both waves, we tested the extent to which their personality changed. We found that the change was minimal, with around 75% of the observations showing at most a one-point change. This reassured us that treating the personality variables time-constant did not substantially bias the results when looking at personality during young adulthood and its association with nest leaving.”

On page 25, when explaining the second limitation of the use of time-invariant personality, we further added:

“Especially for those who left the parental home in their late 20s and reported on the personality measured around age 20, their personality might be different from what was captured earlier.”

3. Comments: 

Furthermore, even though I am not familiar with the analytical strategy that was used, it seems to be straightforward to me. The figures are quite helpful. The analyses, however, seem to rely on the linearity assumption within and across levels as well as for main and interaction effects. Given the relatively good sample size, would it not be possible to model nonlinear effects, additionally? This may help to unravel further interesting trends that may or may not in line with your hypotheses. I was thinking about response surface analyses with the siblings’ trait scores as the two predictors for the similarity of departure. This would be a completely other approach, but would allow linear and nonlinear as well as interaction effects. Anyway, I think that it might be worth considering nonlinearity to get more out of the data.

Response: 

Thank you for the suggestion of trying out non-linear effects. We tested the quadratic terms of all Big Five traits. Among them, we found that only the quadratic term of at-risk children’s conscientiousness made an additional contribution: if a sibling with low conscientiousness had left, an at-risk child was more likely to leave if he/she was conscientious. Also, if a sibling who was highly conscientious had left, an at-risk child scoring low in conscientiousness was more prone to leaving. 

The results can be found in Fig A1 of Additional Analysis 5 in the Appendices. However, this increased flexibility of an already complex model also heightens the risk of overfitting, thus especially this finding requires replication before firm conclusions can be made. As such, we did not elaborate too much on it in the main text. On page 21 of the main text, we added: 

“Finally, we explored whether there was any non-linear effects of the Big Five traits. The results can be found in Fig A1 of additional analysis in the Appendices.”

 On page 7 of the Appendices, we added:

“In this additional analysis, we tested the non-linear effects of all Big Five traits, by adding the quadratic terms. We found that only the quadratic term of at-risk children’s conscientiousness made an additional contribution (see Fig. A1): if a sibling with low conscientiousness had left, an at-risk child was more likely to leave if he/she was conscientious. Also, if a sibling who was highly conscientious had left, an at-risk child scoring low in conscientiousness was more prone to leaving. However, this increased flexibility of an already complex model also heightens the risk of overfitting, thus especially this finding requires replication before firm conclusions can be made.”

4. Comments: 

In sum, the study is interesting and should be published. However, the authors should revise their paper and do a better job in really combing and combining psychological and sociological perspectives by taking genetic factors into account. They should also do a better job in embedding their study into the broad literature on personality development.

Thanks for asking me to read this interesting study.

I wish the authors good luck with their paper,

Christian Kandler

References

Bleidorn, W., Schwaba, T., Zheng, A., Hopwood, C. J., Sosa, S. S., Roberts, B. W., & Briley, D. A. (2022). Personality stability and change: A meta-analysis of longitudinal studies. Psychological Bulletin, 148(7-8), 588–619. https://doi.org/10.1037/bul0000365

Briley, D. A., Livengood, J., & Derringer, J. (2018). Behavior genetic frameworks of causal reasoning for personality psychology. European Journal of Personality, 32, 202–220. http://dx.doi.org/10.1002/per.2153

Briley, D. A., Livengood, J., Derringer, J., Tucker-Drob, E. M., Fraley, R. C., & Roberts, B. W. (2019). Interpreting behavior genetic models: Seven developmental processes to understand. Behavior Genetics, 49, 196–210. http://dx.doi.org/10.1007/s10519-018-9939-6

Kandler, C., & Papendick, M. (2017). Behavior genetics and personality development: A methodological and meta-analytic review. In J. Specht (Ed.), Personality development across the lifespan (pp. 473–495). San Diego, CA: Elsevier Academic Press. http://dx.doi.org/10.1016/B978-0-12-804674-6.00029-6

Kandler, C., Waaktaar, T., Mõttus, R., Riemann, R., & Torgersen, S. (2019). Unravelling the interplay between genetic and environmental contributions in the unfolding of personality differences from early adolescence to young adulthood. European Journal of Personality, 33,

221–244. http://dx.doi.org/10.1002/per.2189

Vukasovic, T., & Bratko, D. (2015). Heritability of personality: A meta-analysis of behavior genetic studies. Psychological Bulletin, 141, 769–785. http://dx.doi.org/10.1037/bul0000017

---

## [Decision Letter · Decision Letter 1]

10 Apr 2023

Do birds of a feather leave the nest together? The role of sibling personality similarity in transition to adulthood

PONE-D-22-27514R1

Dear Dr. Her,

We’re pleased to inform you that your manuscript has been judged scientifically suitable for publication and will be formally accepted for publication once it meets all outstanding technical requirements.

Kind regards,

Srebrenka Letina, Ph.D.

Academic Editor

PLOS ONE

Additional Editor Comments (optional):

The reviewers agree that the manuscript should be accepted at this stage.

Reviewers' comments:

Reviewer's Responses to Questions

**Comments to the Author**

1. If the authors have adequately addressed your comments raised in a previous round of review and you feel that this manuscript is now acceptable for publication, you may indicate that here to bypass the “Comments to the Author” section, enter your conflict of interest statement in the “Confidential to Editor” section, and submit your "Accept" recommendation.

Reviewer #1: (No Response)

Reviewer #2: All comments have been addressed

2. Is the manuscript technically sound, and do the data support the conclusions?

Reviewer #1: Yes

Reviewer #2: Yes

3. Has the statistical analysis been performed appropriately and rigorously? 

Reviewer #1: Yes

Reviewer #2: Yes

4. Have the authors made all data underlying the findings in their manuscript fully available?

Reviewer #1: Yes

Reviewer #2: Yes

5. Is the manuscript presented in an intelligible fashion and written in standard English?

Reviewer #1: Yes

Reviewer #2: Yes

6. Review Comments to the Author

Reviewer #1: (No Response)

Reviewer #2: The authors did a very good job in revising their paper. They addressed all of my comments and concerns in a very detailed fashion. I have no further concerns and recommend acceptance.

7. PLOS authors have the option to publish the peer review history of their article (what does this mean?). If published, this will include your full peer review and any attached files.

Reviewer #1: No

Reviewer #2: **Yes: **Christian Kandler

---

## [Editor Report · Acceptance letter]

24 Apr 2023

PONE-D-22-27514R1 

Do Birds of a Feather Leave the Nest Together? The Role of Sibling Personality Similarity in the Transition to Adulthood 

Dear Dr. Her:

I'm pleased to inform you that your manuscript has been deemed suitable for publication in PLOS ONE. Congratulations! Your manuscript is now with our production department. 

Kind regards, 

on behalf of

Dr. Srebrenka Letina 

Academic Editor

PLOS ONE